# Circular photogalvanic spectroscopy of Rashba splitting in 2D hybrid organic–inorganic perovskite multiple quantum wells

Xiaojie Liu[1], Ashish Chanana[1,2], Uyen Huynh[1], Fei Xue [3,4], Paul Haney[3], Steve Blair[2], Xiaomei Jiang[5]* & Z.V. Vardeny[1]*

The two-dimensional (2D) Ruddlesden−Popper organic-inorganic halide perovskites such as (2D)-phenethylammonium lead iodide (2D-PEPI) have layered structure that resembles multiple quantum wells (MQW). The heavy atoms in 2D-PEPI contribute a large spin-orbit coupling that influences the electronic band structure. Upon breaking the inversion symmetry, a spin splitting ('Rashba splitting') occurs in the electronic bands. We have studied the spin splitting in 2D-PEPI single crystals using the circular photogalvanic effect (CPGE). We confirm the existence of Rashba splitting at the electronic band extrema of 35±10 meV, and identify the main inversion symmetry breaking direction perpendicular to the MQW planes. The CPGE action spectrum above the bandgap reveals spin-polarized photocurrent generated by ultrafast relaxation of excited photocarriers separated in momentum space. Whereas the helicity dependent photocurrent with below-gap excitation is due to spin-galvanic effect of the ionized spin-polarized excitons, where spin polarization occurs in the spin-split bands due to asymmetric spin-flip.

[1] Department of Physics & Astronomy, University of Utah, Salt Lake City, UT 84112, USA. [2] Department of Electrical Engineering, University of Utah, Salt Lake City, UT 84112, USA. [3] Physical Measurement Laboratory, National Institute of Standards and Technology, Gaithersburg, MD 20899, USA. [4] Institute for Research in Electronics and Applied Physics & Maryland Nanocenter, University of Maryland, College Park, MD 20742, USA. [5] Department of Physics, University of South Florida, Tampa, FL 33620, USA. *email: xjiang@usf.edu; val@physics.utah.edu

In recent years the three-dimensional (3D) hybrid organic–inorganic halide perovskites (HOIP) with the general formula of $AMX_3$, where A is an organic cation such as methylammonium (MA), M a divalent metal cation such as $Pb^{2+}$, and $X^-$ the halide anion such as $I^-$, have been extensively studied[1–3]. The superior optical and electronic properties of these semiconductors have opened a wide range of optoelectronic applications, such as photovoltaic devices, light emitting diodes, and lasers[4–9]. The heavy atoms, namely lead and halogen endow HOIP with large spin-orbit coupling (SOC). In the presence of structural inversion symmetry breaking, the SOC leads to spin-splitting of the continuum bands, a phenomenon known as 'Rashba splitting' (see Fig. 1d)[10–14]. This may open a new avenue of using HOIP in spin-related optoelectronic and spintronic applications[15–17].

The 2D version of the HOIP are the multilayered Ruddlesden-Popper compounds with the general formula of $A_2A'_{n-1}M_nX_{3n+1}$, with the corner-sharing $[MX_6]^{4-}$ octahedra forming the inorganic perovskite layers separated by the bilayer of interdigitated long chain organic cation A[18,19], where n indicates the number of inorganic perovskite layers. When n > 1, a small organic cation A' can also be intercalated in the cube formed by eight corner-sharing $[MX_6]^{4-}$ octahedra. 2D HOIP offer superior stability over the 3D counterparts due to the protection of the hydrophobic organic layer, and tunability owing to the synthesis versatility[18–20]. The 2D HOIP form natural multiple quantum well (MQW) structure in which the inorganic $[MI_6]^{4-}$ layers serve as the potential 'wells' and the organic cation chains are the potential 'wall' (see Fig. 1a)[21,22]. The 2D-phenet hylammonium lead iodide (2D-PEPI), $(C_6H_5(CH_2)_2NH_3)_2PbI_4$, $(n = 1)$ is a model example of such natural MQW. At low temperature (~200 K), its crystal structure is monoclinic (space group C2/m)[23,24]. There is a lack of consensus on the room temperature crystal structure in the literature, with the majority reporting a triclinic structure (space group $\bar{1}$)[19,25,26]. Both C2/m and $P\bar{1}$ space groups are centrosymmetric; however, responses associated with broken inversion symmetry may still occur due to the presence of interfaces/surfaces in the MQW structure[14,27]. Giant Rashba splitting was inferred in a recent spectroscopic study of 2D-PEPI[28] and has since initiated strong interest in this material[25,26,29]. However, a direct observation of a spin-related photocurrent and the symmetry breaking direction are still missing. Furthermore, unlike the well-studied doped III–V semiconductor MQW structures, where free carriers carry the spin photocurrent[30], it is largely unknown what role, if any, the excitons in 2D-PEPI play in the photogeneration of spin current. The circular photogalvanic effect (CPGE) is considered as the most important experiment that verifies the existence of Rashba splitting in the electronic bands[14]. CPGE has been studied in a variety of Rashba type materials including doped GaAs/AlGaAs MQW[31], the polar semiconductor BiTeI[32], 2D transition-metal dichalcogenides[33], and topological insulator[34]. Most recently, the CPGE was studied in bulk methylammonium lead iodide perovskite ($MAPbI_3$) films[35] and single crystal[36]. Here we employ helicity-dependent steady state

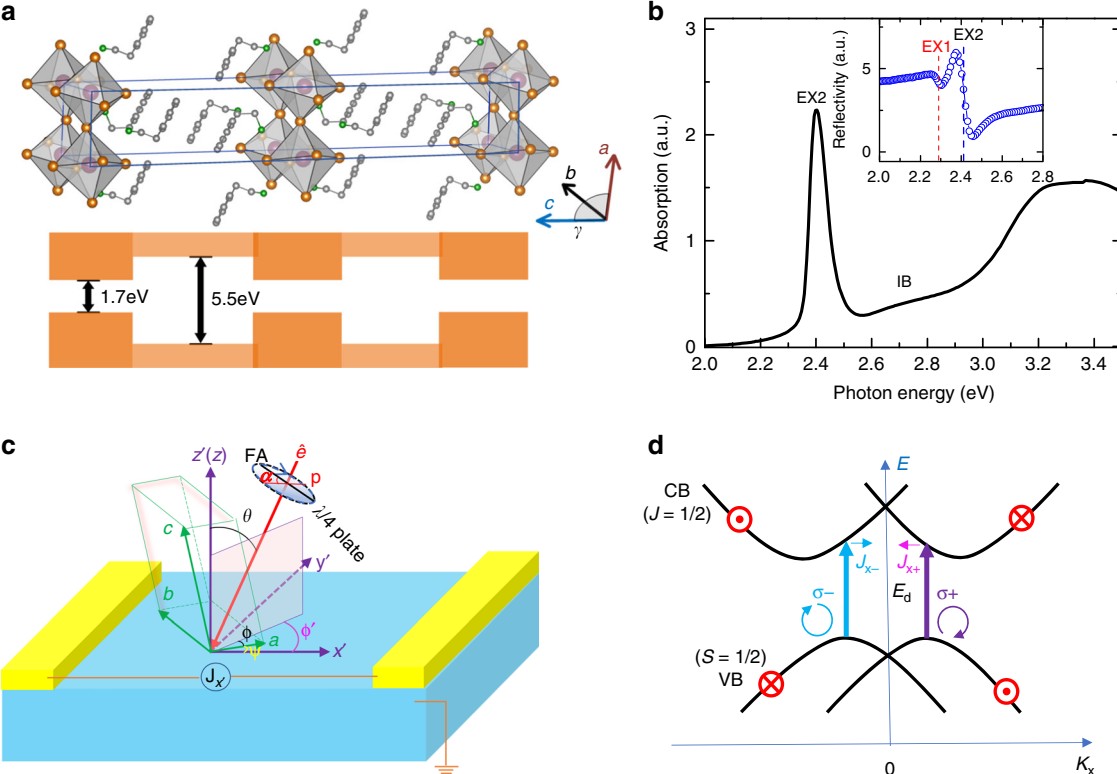

**Fig. 1 Rashba splitting in 2D hybrid organic–inorganic perovskite (2D-PEPI) crystal. a** Schematic of the 2D-PEPI structure with alternating $(C_6H_5C_2H_4NH_3^+)$ and $[PbI_6]^{4-}$ layers, which form natural multiple quantum wells (MQW), where the inorganic layer is the potential 'well' and organic layer is the potential 'wall'. The potential values were taken from ref. [22] The crystal structure is triclinic at room temperature, with growth direction along the c-axis. **b** Room temperature absorption (black line) spectrum of a thin film 2D-PEPI where the exciton (EX) and interband (IB) transitions are denoted. The inset shows the reflectivity spectrum of a single crystal (blue symbol), having two different spectra features. The red(blue) broken line marks the two exciton species (EX1/EX2) observed in the 2D-PEPI crystal. **c** Experimental setup for measuring the PGE using $\lambda/4$ plate; the angles $\alpha$, $\theta$, and $\phi$ are denoted. x' indicates the current flow direction, making an angle, $\psi$ with the crystal a-axis. **d** Schematic diagram of the continuum bands (VB and CB) having Rashba spin splitting, and related optical transitions with circular polarized light. The electron group velocity (and current) change polarity when the light changes helicity. $E_d$ is the direct energy difference between the CB and VB Dirac points.

photocurrent and terahertz (THz) transient emission spectroscopies for studying the CPGE in 2D-PEPI single crystals at room temperature. Our results confirm the existence of Rashba splitting with energy of 35 ± 10 meV at the bands extrema and identify the main inversion symmetry breaking direction to be perpendicular to the MQW planes. The CPGE action spectrum shows two distinctive features that are, respectively, associated with the split interband (IB) transition and polarized exciton (EX) excitation.

## Results

**Continuous-wave (CW) photogalvanic (PGE) currents in 2D-PEPI crystal.** As an introduction to the optical properties of 2D-PEPI, Fig. 1b shows the absorption spectrum of a thin film at room temperature (RT). The absorption spectrum has a pronounced peak at $E_{ex} \approx 2.40$ eV due to the exciton absorption, followed by a monotonic increase related to interband absorption, with an onset at ≈2.57 eV[28]. Other than a small blue shift (~5 meV), the absorption of film and single crystal is essentially the same, with nearly identical interband absorption edge[25]. At lower temperature the exciton band in 2D-PEPI film splits into two excitons, exciton 1 (EX1) and exciton 2 (EX2) that are ~40 meV apart[28,37]. The split of the exciton band in 2D-PEPI can be seen even at RT in single crystal that has less disorder than in film, as evident in the reflectivity spectrum shown in Fig. 1b inset (marked by the two broken lines).

Figure 1c illustrates the experimental geometry used for the helicity-dependent photocurrent measurement; the definition of various axis and angles are given in Fig. 1c caption. We used a quartz $\lambda/4$ plate (QWP) to modulate the light polarization from right circular polarization (RCP, $\sigma+$), to linear polarization (LP), to left circular polarization (LCP, $\sigma-$) by rotating the angle, $\alpha$ between the fast axis of the QWP and the incident light polarization. In addition, the light excitation intensity was modulated at frequency $f = 310$ Hz using a mechanical chopper, and the resulting photocurrent vs. $\alpha$ was measured using a phase sensitive technique (see Methods section in the SI). Importantly, the CPGE response is obtained at zero bias voltage. We note that the degree of circular polarization, $P_{circ}$ varies with the rotation angle $\alpha$ as $P_{circ} = \sin 2\alpha$; namely $P_{circ} = +1$ for RCP (or $\sigma+$) when $\alpha = 45°$, 225°..., $P_{circ} = -1$ for LCP when $\alpha = 135°...$, 135°..., and $P_{circ} = 0$ for LP light, when $\alpha = 90°$, 180°...[30].

In HOIP the conduction band (CB) bottom consists of J = ½ states, whereas the valence band (VB) top has S = ½ states[38]. The optical transition selection rules for in-plane spin polarization require that the component of angular momentum directed along the Rashba effective magnetic field changes by $\Delta m_J = \pm 1$. Taking this into consideration, Fig. 1d shows a schematic in one dimensional $\mathbf{k}$-space. When Rashba splitting occurs in the CB and VB, the absorption of RCP (or $\sigma+$) light results in interband transition (from $m_j = -1/2$ to $+1/2$) between the right branches ($k_x > 0$), whereas the absorption of LCP (or $\sigma-$) light allows transition (from $m_j = +1/2$ to $-1/2$) between the left branches ($k_x < 0$). Therefore the circularly polarized light creates nonequilibrium spin polarization among the two Rashba branches. Through the inverse Rashba-Edelstein effect[39], a CPGE photocurrent, $J_{x+}$ ($J_{x-}$) is generated when the excited-state electrons (holes) relax to the CB (VB) minimum (maximum). Since the group velocity $d\varepsilon/dk_x$ of electrons are opposite in direction and equal in magnitude along $k_x$, $J_{x+}$, and $J_{x-}$ are opposite in directions (and similarly for the holes in the splitted VB). We note that the CPGE current is ultrafast via the momentum scattering that randomizes $\mathbf{k}$ and, in turn also $\mathbf{S}$, since $\mathbf{k}$ and $\mathbf{S}$ vectors are locked together in a Rashba-splitting situation. Therefore, there is associated terahertz (THz) radiation from the resulting time-varying current[35]. At steady state conditions,

the continuous generation of spin-polarized free carriers leads to a continuous-wave (cw) photocurrent that is sensitive to the polarization status of excitation light[30,35,36]. Since the group velocity of free carriers at the bottom (top) of CB (VB) is zero (Fig. 1d), which consequently results in null photocurrent in the absence of bias, the CPGE is zero at/below the IB absorption onset. Importantly, CPGE does not exist if the CB and VB possess spin ½ Kramer's degeneracy, since the photoinduced $\mathbf{k}$-vector of both LCP and RCP is the same, and thus the group velocity (and photocurrent) does not change direction when the angle $\alpha$ is changed via the QWP rotation.

In addition to the CPGE current associated with circularly polarized light, there is also spin-independent photocurrent when the excitation light is linearly polarized (LP, for $\alpha = n(\pi/2)$)[30] This is known as linear photogalvanic effect (LPGE). Through a 360° rotation of the QWP, the incident light polarization cycles through LP, RCP, LP, LCP, and therefore the measured photogalvanic effect (PGE) current, $J_{x'}(\alpha)$ contains both CPGE and LPGE current given by:

$$J_{x'}(\alpha) = C1 \sin(2\alpha) + C2 \cos(2\alpha) + L1\sin(4\alpha) + L2\cos(4\alpha) + D$$

(1)

Where $x'$ indicates the measurement direction in Fig. 1c. The $4\alpha$ terms describe LPGE, the $2\alpha$ terms stand for CPGE, and $D$ represents a polarization-independent offset that originates from other effects such as photothermal, photovoltaic response, or photo-Dember effect[30,35,36,40]. The cosine term, $C2$ is mainly due to misalignment (about 1~ 3 degree, in our case) between the fast axis of the $\lambda/4$ waveplate and the incident light polarization, so that $C2 \ll C1$. The relative strengths of LPGE parameter pairs ($L1$, $L2$) depend on several experimental conditions and symmetry tensor (see SM11 in SI, and Auxiliary Supplementary Materials). In our measurements we found that $L1 \ll L2$ (see Supplementary Table 2 and 4 in **SI**). We note that the coefficients $C1$ (for CPGE) and $L2$ (for LPGE) depend on the incident angle, $\theta$ and azimuthal angle, $\phi'$ (defined in Fig. 1c), as detailed in SM5&9 in **SI**. In general, $C1$ and $L2$ are related to a second order tensor, $\gamma_{ij}$, and third order tensor, $\chi_{ijk}$, respectively[30]. The point group symmetry for the triclinic 2D-PEPI is $C_i$[19,25,26]. Our calculation shows that the inversion symmetry is broken along z-axis (out-of-plane) and the crystal b-axis (in-plane) (Fig. 1c). There are three non-zero components for $\gamma_{ij}$ tensor in $C_i$, two are in-plane ($\gamma_{xy}$ and $\gamma_{yx}$) and one is out-of-plane, $\gamma_{xz}$. For $\chi_{ijk}$, there are five non-zero components: $\chi_{xxy}, \chi_{xxz}, \chi_{yxx}, \chi_{yyy}, \chi_{yyz}$. (see SM11 in SI, and Auxiliary Supplementary Materials for details).

Figure 2a shows the room temperature PGE($\alpha$) response [$J_{x'}(\alpha)$] in 2D-PEPI single crystal at steady state conditions and incident angle $\theta = 35°$, using a xenon lamp excitation at 2.30 eV for EX generation, and 2.64 eV for IB excitation, respectively, The $C1$, $L2$, and $D$ parameters are obtained from fitting using Equ.(1), and seen as colored bars (Fig. 2a inset). This procedure was repeated for various incident ($\theta$) and azimuthal ($\phi'$) angles, as well as excitation energies ($\omega$), in order to obtain the complete PGE response of the 2D-PEPI crystal. As is clearly seen, both CPGE ($C1$) and LPGE ($L2$) have opposite signs for resonant excitation at the EX and IB transitions. This difference in PGE is further illustrated in Fig. 2b, where the CPGE action spectrum, $C1(\omega)$ of the 2D-PEPI crystal is displayed (black symbol). It is seen that $C1$ ($\omega$) has two spectral regions that are separated at ~2.40 eV; $\hbar\omega$ <2.40 eV for photogenerated excitons and $\hbar\omega$ >2.40 eV for excited-state photocarriers generated via IB transition. The CPGE action spectrum for the IB transition (CPGE-IB) is very broad with a maximum at ≈2.52 ± 0.03 eV, whereas the CPGE spectrum related to EX transition (CPGE-EX) consists of a sharp peak at 2.34 ± 0.02 eV and negative valley at 2.30 ± 0.02 eV, with a split of

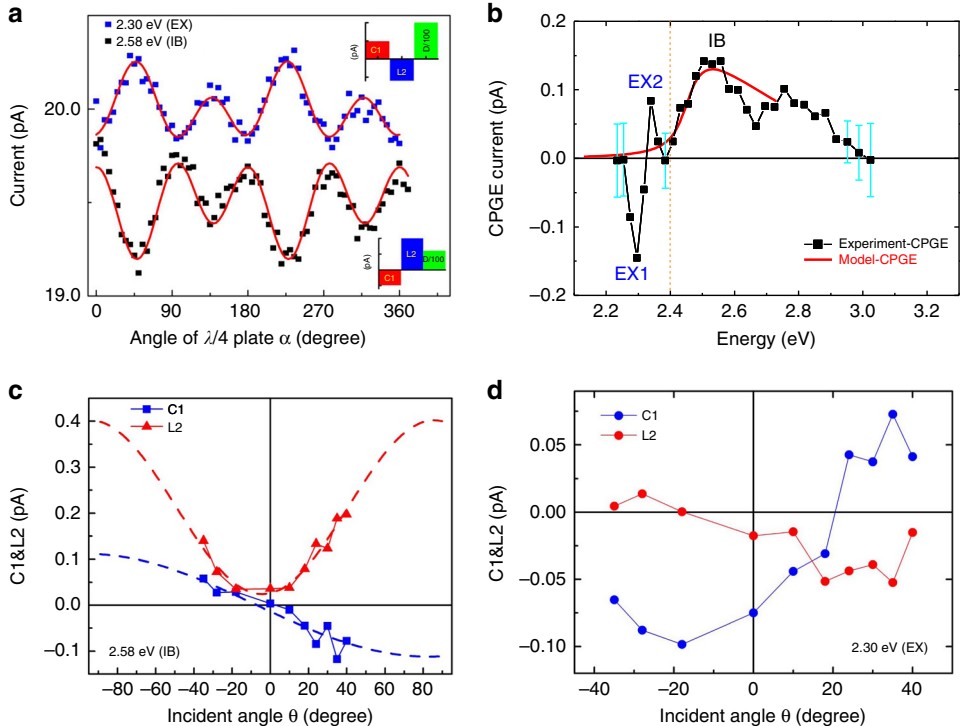

**Fig. 2 Continuous-wave (CW) photogalvanic (PGE) currents in 2D-PEPI crystal. a** The room temperature photogalvanic current in 2D-PEPI crystal vs. $\lambda/4$ plate rotation angle, $\alpha$ measured at $\theta = 30°$, excited at resonance with the exciton (blue squares) and interband (black squares) that is shifted vertically for clarity. The red lines through the data points are fits using Eq. (1) with fitting parameters C1, D, and L2 shown in the insets. **b** The CPGE (C1) action spectrum of 2D-PEPI crystal (black squares); the red line through the data points is a fit using a four bands model (see text). The error bars (cyan, s.e.m.) indicate the uncertainty of special data points close to zero. $C1(\omega)$ spectrum is divided by a vertical broken line into two spectral ranges, exciton (EX) and interband (IB). The two exciton species are labeled as EX1 & EX2. **c** The CPGE amplitude, C1 (blue squares) and LPGE amplitude, L2 (red triangles) vs. the incident angle, $\theta$ at resonant excitation with the interband (IB, at 2.58 eV). The broken lines are fittings (see text). **d** The incident angle $\theta$ dependence of C1 (blue circles) and L2 (red circles) at resonant excitation with the exciton (EX1, at 2.30 eV).

~40 meV, reminiscence of EX1 and EX2 in the reflectivity spectrum of 2D-PEPI (Fig. 1b inset). The uncertainty comes from the reduced spectral resolution of the setup (see SM2 in **SI**). In addition, CPGE-IB has associated terahertz (THz) emission, whereas THz emission is not observed when resonantly exciting the excitons with similar photon density ($\sim 4 \times 10^{17}$ photons cm$^{-3}$) (see Fig. 3b, c below). This shows that the CPGE-IB is in fact ultrafast, whereas the CPGE-EX is not. This indicates that the CPGE-EX current decays (or generated) much slower than sub-picosecond timescale (limit of our setup), and therefore it does not generate THz emission (see below).

Figure 2b also shows the calculated CPGE-IB action spectrum (red line) using a model system consisting of $J = \frac{1}{2}$ states for the CB and $S = \frac{1}{2}$ states for the VB. The model describes the band edge states in 2D-PEPI crystal and accommodates a Rashba term (see SM12 in **SI**). The key result from our model calculation is that the CPGE-IB spectrum is peaked at energy, $E_d$ that corresponds to the direct transition at $k = 0$ between the Dirac points in the CB and VB (see Fig. 1d). CPGE-IB has a threshold energy at $E_d - (m_c^* + m_v^*)(\alpha_c + \alpha_v)^2/(2\hbar^2)$, where $\alpha_{c(v)}$ is the Rashba coefficient of the CB (VB), and $m_{c(v)}^*$ is the effective mass of the CB (VB). Note that the Rashba-splitting energy, $E_R$ corresponds to the energy difference between the threshold and maximum of the CPGE-IB action spectrum, i.e., $E_R = (m_c^* + m_v^*)(\alpha_c + \alpha_v)^2/(2\hbar^2)$. Figure 2b shows that the experimental CPGE-IB action spectrum is well reproduced by this model, using Rashba splitting of $35 \pm 10$ meV and broadening parameters of $30 \pm 10$ meV (details in SM12 in **SI**). The good agreement with the experimental data validates the Rashba splitting in 2D-PEPI continuum bands.

In order to experimentally identify the symmetry breaking axis in 2D-PEPI crystal that leads to the Rashba splitting, we measured the incident angle ($\theta$) dependence of the PGE($\theta$) response at 2.58 eV (IB) (Fig. 2c) and 2.30 eV (EX1) (Fig. 2d) at azimuthal angle $\phi' = 90°$, For the CPGE-IB in this special case $C1(\theta) = A\sin(\theta) + B\cos(\theta)$, whereas $L2(\theta) = A'\sin(2\theta) + B'\cos(2\theta) + G$. A, B, A', B' and G are fitting parameters with details given in SM11 in **SI**. The dashed lines in Fig. 2c show the fitting results. The satisfactory fitting for $L2(\theta)$ with this model proves that the break of inversion symmetry exists along the z-axis (out-of-plane) and crystal b-axis (within the [PbI$_6$]$^{4-}$ plane) (Fig. 1c). The fitting for $C1(\theta)$ is not as good and yields near zero $\gamma_{xz}$, meaning that CPGE mostly results from inversion symmetry breaking along the out-of-plane z-direction. Indeed, C1 changes sign when $\theta$ is reversed, and passes through zero at $\theta = 0°$. This is due to the coincident line-up of the photocurrent direction ($x'$) in this device with the crystal b-axis (within the [PbI$_6$]$^{4-}$ plane). Since the inversion symmetry is also broken along b-axis, a null CPGE is expected if measured in this direction[30]. Non-zero C1 was observed at normal incidence when the measurement direction was not lined up with b-axis (see Supplementary Table 4). Details of the fitting analysis can be found in SM11 in **SI**.

**Terahertz (THz) emission due to ultrafast photogalvanic (PGE) currents in 2D-PEPI crystal.** For demonstrating the instantaneous generation of the PGE response at the IB excitation, we have also used transient THz emission spectroscopy to complement the steady state measurements. Figure 3a is a schematic illustration of the experimental setup for the helicity dependent

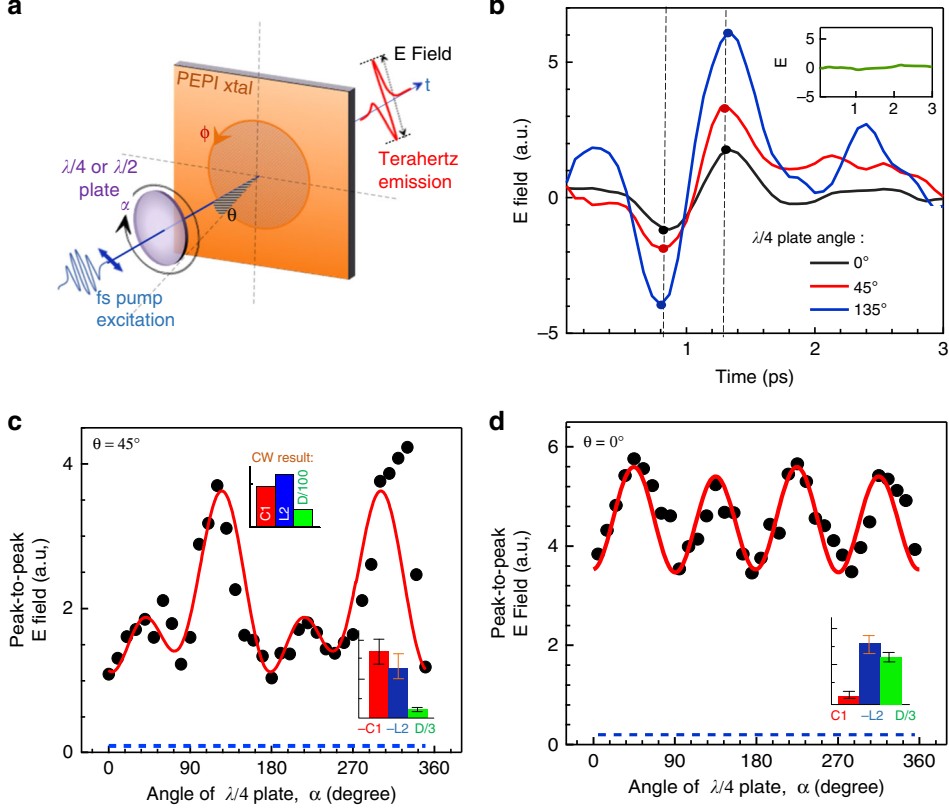

**Fig. 3 Terahertz (THz) emission due to ultrafast photogalvanic (PGE) currents in 2D-PEPI crystal. a** schematic illustration of the experimental setup for the THz emission measurements. The angles α, θ, and φ are denoted. **b** THz time-domain emission waveforms measured at λ/4 waveplate angle, α of 0°, 45° and 135° as denoted, that correspond to linear polarized (LP), right circularly polarized (RCP) and left circularly polarized (LCP) light. The two dashed black lines mark the times where positive and negative peaks of the terahertz emission were determined. The peak-to-peak values are determined from the addition of the absolute positive and negative values as marked by the dots. The inset shows a null signal obtained when resonantly excited at the exciton band (2.34 eV). **c** Terahertz field emission vs. the rotation angle, α, photogenerated using 3.06 eV pulsed excitation at θ = 45°. The red line through the black data points is a fit using Eq. (1). The inset shows the obtained relative values of the coefficients C1, L2, and D; **d** Similar measurements as in panel **c**, for incident angle, θ = 0°. The fit using Eq. (1) shows majority contribution from L2. The blue dashed line marks the noise level corresponding to zero emission field.

THz emission measurement. This setup is similar to the one used for the steady state measurements (see Fig. 1c), except that the crystal is devoid of gold electrodes, hence eliminating related field effect artifacts. The 2D-PEPI crystal is excited with ~50 femtosecond (fs) second-harmonic pulses at 3.06 eV using an amplified Ti-sapphire laser operating at a center wavelength of 810 nm, and the THz emission is collected in transmission geometry (see SM9 in **SI**). Figure 3b shows the time-domain waveforms at three different rotation angles (α) of the QWP measured at normal incidence. The detection electro-optic crystal and a wire grid polarizer were used to measure the emitted field in the y'-direction (with angle ψ to the b-axis, see Fig. 1c). The absence of sign reversal for the emission field between RCP (α = 45°) and LCP (α = 135°) can be explained by the presence of in-plane inversion symmetry breaking, similar as observed in ref. [30], consistent with the measurement in Fig. 2c.

Figure 3c shows the THz field emission vs. α at incident angle θ = 45°. The data points for the **E**-Field are extracted as 'peak-to-peak' values in the THz emission signal (see the broken lines in Fig. 3b). The peak-to-peak value of the **E**-field corresponds to the amplitude of the PGE current, J, as defined in the phenomenological model described in Eq. (1). The red line is a fit using Eq. (1). The lower inset shows the obtained relative values of the coefficients C1, L2, and D. It is clearly seen that the THz emission intensity depends on the pulsed light polarization helicity, similar to the steady state results in Fig. 2a, and has a strong sin(2α)

component that originates from CPGE. Ultrafast current is known to result in pulsed THz emission that corresponds to subpicosecond electric field, and the THz emission is proportional to the time derivative of the photocurrent. Since the duration of the pulse excitation is ~50 fs in our THz setup, the relatively strong THz emission here indicates that the polarized photocurrent is very short lived. This is consistent with the process of PGE current that is estimated to happen in the femtosecond timedomain, associated with momentum scattering time of the material, as described in Fig. 1d. We note that THz emission associated with fast PGE current has been presented in previous studies[35,41,42]. We also conclude that the THz emission is unlikely to originate from optical phonons, as such process would have a different characteristic transient; namely long-lived periodically modulated signal[43,44] rather than a single cycle emission as measured in Fig. 3b. In fact, we measured the optical phonon modes using terahertz transmission through the crystal w.r.t. to reference substrate. Phonons in 2D-PEPI can be observed at 0.78 THz and 1.6 THz, respectively (see Supplementary Fig. 12 in SI); probably associated with Pb-I-Pb rocking vibration and Pb-I stretching modes[43]. Moreover, the helicity dependence of the THz emission points to a spin-dependent process, which is also present in CW measurements, rather than simply due to phonons.

We also measured substantial ultrafast LPGE component (see Supplementary Fig. 13) at normal incidence indicating that this

linear PGE effect is due to 'shift current', caused by the displacement of the electron charge center upon undergoing a transition from the VB to CB[45] (see SM9 in SI). Figure 3d shows similar measurements as in Fig. 3c but at normal incidence ($\theta = 0°$). The extracted parameters show much smaller contribution of C1 at normal incidence, in agreement with the CW measurements (Fig. 2c). We note in passing that an important difference exists between the THz and steady state measurements, which is the much-reduced contribution of the THz emission associated with the background current D (see inset of Fig. 3c). It is conceivable that D in THz emission mainly comes from the (ultrafast) photo-Dember effect[44], whereas in CW excitation slow processes, such as photothermal effect dominate[40].

In contrast to the spectral response of CPGE-IB, the spectrum of CPGE-EX has a completely different line shape (Fig. 2b, left of the broken line). Here the action spectrum comprises of a derivative-like feature with a negative valley at 2.30 eV and peak at 2.34 eV; however it is not the first derivative of the absorption. Furthermore, upon 180° rotation of the device the derivative-like feature converges into a single band (see Supplementary Fig. 3 in SI). Also, as shown in Fig. 2a, the PGE($\alpha$) response at the exciton photon energy is very different from that at the interband. In addition, the incident angle $\theta$−dependence of both CPGE (C1) and LPGE (L2) at the exciton energy (Fig. 2d) are also very different from those at the interband (Fig. 2c). In fact, for the exciton excitation, $C1 \neq 0$ at $\theta = 0°$, and L2 is almost zero at negative $\theta$ angle. Importantly, no THz emission related to the CPGE-EX was observed (see Fig. 3b inset), indicating a slower dynamic of the helicity-dependent photocurrent generated from the exciton dissociation. Furthermore, the sense of spin is opposite at the valley and peak of the CPGE-EX spectrum. We thus conclude that the spin-dependent photocurrent associated with the photogenerated excitons is a feature that cannot be explained by the CPGE traditional band model[30,35,36].

Clearly the excitons in 2D-PEPI substantially contribute to the photocarriers density (see photoconductivity action spectrum in SI, Supplementary Fig. 4). Although the PGE current was measured at zero bias, at steady state there is still a small electric field (estimated to be about 500 V/cm, see SM5 in SI) within the device that originates from the photothermal effect due to light-induced temperature gradient across the device and/or photovoltaic effect from slight asymmetry between the two Au electrodes[35,40]. This weak electric field contributes to the DC offset current D (see MM5 in SI). However, based on our calculation, this weak electric field is not strong enough to dissociate the excitons in 2D-PEPI because of the large exciton binding energy here (>250 meV)[24,29,37]. Yet, as shown in ref. [46] efficient exciton dissociation may occur at native defects in the crystal. We also note that the exciton CPGE occurs below the exciton main absorption, and that the CPGE (C1) current polarity depends on the incident angle $\theta$ (Fig. 2d).

One possible mechanism to explain these puzzling results is that the helicity-dependent photocurrent at the exciton band is in fact due to the spin-galvanic effect (SGE) rather than the CPGE. In this scenario the spin angular momentum of the impinging light is conserved during the absorption by the exciton. To verify this assumption we have measured a transient circular polarization memory at the exciton level at RT using the transient polarized photoinduced absorption technique at 537 nm with 150 fs time resolution. In this method we set the pump beam polarization at a fix circular polarization, whereas the circular polarization of the probe beam was modulated between same circular polarization or opposite polarization to that of the pump beam (see SM10 in **S.I**. for detail). In this method only the difference between the same or opposite pump-probe circular polarizations is measured. Firstly, we found that there is 'circular

polarization memory' for the exciton, in which the photoinduced absorption is larger when the pump-probe have same polarization compared to that of pump-probe with opposite polarization (see Supplementary Fig. 16 in SM10, **SI**). This shows that the excitons are spin polarized following excitation by a circularly polarized pump. Secondly, we measured the RT circular polarization lifetime, or spin relaxation time to be ~4.5 ps on the average (see Supplementary Fig. 17 in SM10, **SI**). Subsequently, some of the spin-polarized excitons dissociate into spin-polarized electron-hole pairs that contribute to the photocurrent. The exciton dissociation process may be via edge states, or other native defects in the 2D-PEPI crystal. Since the continuum bands in 2D-PEPI are spin splitted due to the Rashba interaction, therefore the exciton-related spin-polarized carriers preferentially occupy one spin sub-band over the other, depending on the light beam helicity. This non-equilibrium spin occupancy causes asymmetric spin-flipping between the two spin sub-bands and results in a current flow in the MQW plane. This situation has been known in the literature as spin-galvanic effect (SGE)[47,48]. A similar situation occurs in the Rashba-Edelstein effect upon spin injection from a ferromagnet electrode[49], except that in our case the spin injection occurs by optical means. The SGE in our case is not ultrafast, since it takes some time for the excitons to dissociate at native defects in the crystal. In addition, the spin relaxation time is not in the sub-ps time-domain. These explain the lack of THz emission due to the SGE of the excitons in 2D-PEPI.

## Discussion

Using circularly polarized light excitation we obtained steady state spin-dependent photocurrent and ultrafast terahertz (THz) emission, which verify the existence of CPGE in 2D-PEPI single crystal multiple quantum wells (MQW). The circular photo-galvanic effect (CPGE) action spectrum contains two distinct features that are due to excitons and free carriers, respectively. The CPGE at the interband excitation is a 'smoking gun' proof for Rashba splitting in the continuum bands of 2D-PEPI, which is caused by the large SOC and structural inversion symmetry breaking. We found that the main axis of inversion symmetry breaking is perpendicular to the MQW planes, but there is also small contribution from in-plane inversion asymmetry. In contrast, the spin-dependent photocurrent upon exciton excitation is caused by spin-galvanic effect, which also proves the occurrence of Rashba splitting in 2D-PEPI. Our findings highlight the importance of excitons for helicity-dependent photocurrent in 2D perovskites MQW, a topic that has not been properly dealt with in the well-established CPGE theory in semiconductor MQW.

## Methods

**Samples preparation**. PbI$_2$, R-NH$_3$I (where R is C$_6$H$_5$C$_2$H$_4$), N,N'-dimethylformamide (DMF), g-butyrolactone (GBL), and dichloromethane (DCM) were purchased from Sigma–Aldrich Corporation. All the materials were used as received without further purification.

All samples were fabricated in a nitrogen-filled glove box with oxygen and moisture levels of <1 part per million. We have grown the 2D hybrid perovskite (2D-PEPI) single crystals on cleaned quartz substrates using the Anti-solvent Vapor-assisted Crystallization (AVC) method as in ref. [1] The 2D-PEPI crystals were used for the following measurements: photoluminescence (PL) spectrum, terahertz emission spectroscopy, XRD and SEM microscopy. For the device used in continuous-wave (CW) PGE measurement, two 70 nm thick gold electrodes were deposited onto the crystal by e-beam evaporation through a shadow mask in a glove-box-integrated vacuum deposition chamber (Angstrom Engineering), which had a base pressure of $3 \times 10^{-8}$ torr ($\approx 4 \times 10^{-6}$ Pa). The gap between electrodes was 0.5 mm.

For the solution used to create 2D-PEPI film, we mixed R-NH$_3$I and PbI$_2$ in a 2:1 molar ratio in DMF solvent to form a solution with a concentration of 0.2 mol/ml. This solution was stirred overnight at 60 °C on a hotplate before using. Subsequently the solution was spin-coated on an oxygen plasma–pretreated glass substrate at 314 rad/s and 90 s to form 100 nm thick film; the obtained film was subsequently annealed at 100 °C for 30 min. We used this film for the optical density measurement.

**Continuous-wave (CW) PGE measurement**. CW diode lasers that operate at wavelengths of 405, 447, 486, 520, and 532 nm, respectively, were used to excite the 2D-PEPI single crystal between the two gold electrodes of the device. The laser beam with a diameter of 0.25–0.45 mm was focused exactly at the center between the two electrodes to minimize the effects caused by electrode asymmetry. In these measurements, the laser power was reduced to 45 μW, with a diameter of 0.35 mm, so that the light intensity was 31 mW/cm². For measuring the CPGE action spectrum, we also used as a pump excitation an incandescent light source from a xenon lamp, which was dispersed through a monochromator. Roughly 25% of the light beam was focused on the active area of the device, with an area 0.5 × 0.75 mm, with an intensity of 8.0 mW/cm². Due to the very low intensity of Xenon lamp, we use full-slit width of the monochromator to ensure the needed intensity for measurable signal.

**Photoconductivity action spectrum**. In this measurement, the incandescent light from a Xe lamp, which was dispersed through a monochromator, was used to excite the same device used in the CPGE measurement. We also measured the conductivity by sweeping the voltage applied to the device with a Keithley 238 multimeter. The voltage was swept in a symmetrical way (0 V to −5 V, +5 V to −5 V, −5V to 0 V). The photoconductivity was then subtracted by linearly fitting the I–V curve from +5 V to −5 V. This procedure was adopted from ref. [36].

**Terahertz emission measurements**. Terahertz emission from 2D-PEPI crystals was measured by an electro-optic sampling technique using standard time-domain spectroscopy configuration. The samples were excited by 0.25 μJ pulses at 405 nm generated using type-I BBO crystal pumped with 810 nm pulses from Ti-Sapphire regen-amplified laser system at 1 KHz repletion rate. 2D-PEPI crystals on Quartz substrate were excited from the quartz side. The emitted terahertz radiation due to photo-excited carriers was collected by 2 parabolic mirrors and focused on to 0.5 mm thick electro-optic ZnTe < 110 > crystal. The terahertz field pulse signal was measured as the change in polarization of the probe beam induced as a result of electro-optic sampling technique, as measured by a Wollaston prism and a set of balanced silicon detectors using lock-in technique. We note that the measured bandwidth of the emitted signal detection technique is limited by the detection crystal. To measure the polarization dependent terahertz field, the sample was mounted on a rotation stage and the excitation beam was modulated using λ/2 and λ/4 plate. In addition, a wire grid polarizer was placed in the collimated beam path between the two parabolic mirrors, to allow detection of polarization emitted terahertz field.

**Optical characterizations**. All optical measurements were done at room temperature in air. The absorption (or optical density) was measured using a UV/Vis spectrometer (Olis). For the photoluminescence (PL) measurement, a 2D-PEPI single crystal was excited using a 30 mW CW laser at 486 nm. The reflectivity spectrum from 2D-PEPI crystal was measured using Woollam variable angle spectroscopic ellipsometer (VASE) on large area crystals grown on quartz substrates with average thickness 8–13 μm. The equipment has built-in setup for reflectivity measurement of s-polarized and p-polarized light.

## Data availability
The data that support the findings of this study are available from the corresponding author upon reasonable request.

## Code availability
We have included the original code for calculation of symmetry tensor as additional supporting material (auxiliary supporting material). The algorithm for band model calculation are available from the corresponding author upon reasonable request.

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

## Acknowledgements
The CW CPGE measurements were supported by the Department of Energy Office of Science, Grant DESC 0014579. The crystal growth facility, the optical and morphological characterizations, and the transient circular polarization measurements were supported by the Center for Hybrid Organic-Inorganic Semiconductors for Energy (CHOISE), an Energy Frontier Research Center funded by the Office of Basic Energy Sciences, Office of Science within the US Department of Energy. The CPGE THz emission studies were supported by the NSF grant EECS 1810096. F.X. acknowledges support under the Cooperative Research Agreement between the University of Maryland and the National Institute of Standards and Technology Physical Measurement Laboratory, Award 70NANB14H209, through the University of Maryland. We also thank Dr. Pete Sercel for fruitful discussion.

## Author contributions
X.L. synthesized and grew the 2D-PEPI crystals and fabricated the devices used in this work. X.L. also conducted the CW CPGE measurements and optical characterizations (absorption, photoluminescence, and photoconductivity). A.C. and S.B. conducted the THz measurements; A.C. also did all structural characterizations (XRD, SEM, AFM). X.L. and A.C. conducted the reflectivity measurement. U.H did the transient circularly polarized pump and probe measurement. F.X. and P.M.H. did the modeling and theoretical calculation of symmetry tensors and CPGE action spectrum. P.M.H also provided software programming. X.J and Z.V.V are the correspondent authors of this work. This has included: conceptualization of the study; design and guidance of experiments; and analysis of the data. X.J. wrote the original draft with input from all authors, and Z.V.V. and P.M.H. edited the consequent versions of the manuscript.

## Competing interests
The authors declare no competing interests.

## Additional information

**Peer Review Information** *Nature Communications* thanks Eugenius Ivchenko, Mikael Kepenekian and Daniel Niesner for their contribution to the peer review of this work. Peer reviewer reports are available.

