## [Peer Review File · Nature Communications]

Reviewers' comments:

Reviewer #1 (Remarks to the Author):

Comments on „Circular photogalvanic spectroscopy of Rashba coupling in 2D hybrid organic-inorganic perovskite quantum wells“ by Xiaojie Liu et al.

The authors report the observation of a circular photogalvanic effect, and of optically driven spin currents, respectively, in a layered organic-inorganic perovskite-like Ruddlesden-Popper compound. To the best of my knowledge, it is the first time the effect is observed in this class of materials. The authors do not only give direct prove of a significant Rashba effect in this layered organic-inorganic compound, but thanks to the high quality of the data and a careful quantitative analysis they also draw conclusion about the symmetry of the crystal underlying the effect. The paper is overall well written. The results are of interest not only to the scientific community working with organic-inorganic perovskites, but more generally to researchers with an interest in spin effects in semiconductors. In this sense, the manuscript absolutely fulfills the standards of Nature Communications. However, I have several questions concerning the experimental details and the interpretation of the data, see below. If those can be fully addressed, I will recommend the manuscript for publication.

Detailed comments:

- The excitonic spectrum consists of two peaks. The one at ~ 2.4 eV is dominant in the reflectivity measurements on crystals (S13, S15) and in the transmission experiments on thin films (Fig. 1B, red curve). The one at 2.3 eV is prominent in the photocurrent measurements on single crystals (Fig. 1B, black symbols) as well as in the CPE action spectrum of the single crystals (Fig. 2B), with the high-energy feature being almost absent in these measurements. The authors conclude that the 2.4 eV feature is intrinsic to the material and assign the low-energy peak to defects. They also conclude that the CPE action spectrum is not the derivative of the absorption spectrum. I think that the comparison between the CPE spectrum of the crystal and the absorption spectrum of the film (as well as to the reflectivity data) is problematic. Crystal and thin film (surface which is also measured in reflectivity, edges of sheets, internal interfaces) could differ in their optical and electronic properties because of differences in their chemical composition, band bendings etc. The two transitions should be discussed carefully.

Photoluminescence experiments on thin films and single crystals, similar to the ones in Lingling Mao's "Hybrid Dion-Jacobson 2D Lead Iodide Perovskites" seem simple and might help with the interpretation of the transition. At present, only a single photoluminescence spectrum is given (Fig. 1B). It is assigned to a crystal in the figure caption and to a thin film in the main text.

- While the CPE experiments and their symmetry analysis give direct evidence of the Rashba splitting and information about its directionality, the interpretation of the Terahertz experiments seems less straightforward to me. What is the origin of the THz emission? That "THz emission is proportional to the time derivative of the photocurrent" does not seem obvious to me. Are the authors referring to the Bremsstrahlung emitted as the electrons relax from the initially optically excited state (with finite group velocity) do the band minimum (with group velocity zero)? Why can the signal not result from optical phonons generated during the relaxation of carriers to the band edge? This would naturally explain that for the excitonic

transition no THz emission is observed, as the excess energy is insufficient for optical phonon generation. What is the origin of the ~ 1 THz periodic modulation of the signal? How are those periodic modulations consistent with the interpretation of the signal as a change in current? How are the data points in Fig. 3C, 3D extracted from the data in 3B – at what time delay?

- The quality of the writing in the abstract needs to be improved to match the one in the main text: continuum bands \rightarrow bands (are continuous by definition), existence of Rashba splitting of 35 meV \rightarrow existence of a Rashba splitting of 35 meV, CPE at interband \rightarrow CPE associated with interband transitions, spin-polarized hot photocarriers whose spins are split in momentum space \rightarrow ... Also, on “2D-PEPI is an exemplary hybrid organic-inorganic halide perovskite semiconductor”: In the main text, the perovskite and the Ruddlesdon-Popper structures are properly distinguished. This might also make sense here.

- Line 40: „PbI layers with n the number of inorganic PbI layers“, line 43: “inorganic PbI well and organic spacer wall”: in general, the quantum wells are not inorganic, but formed by a hybrid organic-inorganic structure, as the authors state correctly in the first sentence of the paragraph. The wording “inorganic PbI layers” is a little confusing, since the reader expect (from abstract and introduction) quantum wells with a perovskite-like structure, rather than a metal halogenide salt. Please use more specific wording.

- I believe the authors use the term “hot electrons” for the electronic state created by the optical excitation. In the literature, in contrast, the term is used for an electronic system that has reached quasi-equilibrium (i. e., electrons have relaxed to the band edges) and can be described by an electronic temperature (therefore the term “hot”) which is different from (and typically much higher than) the one of the phonon bath. I suggest that the authors discriminate clearly between the optically excited state and the electronic system in quasi-equilibrium. This relates to line 94, where the authors state that the hot carrier relaxation/thermalization time determines the decay of the CPGE current. I would rather think that it is the momentum scattering and randomization time which dominates the CPGE decay, in line with a publication by one of the authors (Appl. Phys. Lett. 109, 193903 (2016)).

- lines 42/43: multiple quantum wells structure \rightarrow multiple quantum well structures

- line 49: a/the spin-related photocurrent

- line 56/57: “PL spectra of PEPI thin film” inconsistent with figure caption that claims that PL spectrum of a crystal is shown. Also, there is only one spectrum in the figure.

- line 59/60: “PL spectrum has a dominant band at 2.35 eV that is associated with a large exciton binding energy > 200 meV” band \rightarrow peak; the peak is associated with the exciton (not in any obvious way with its binding energy)

- line 67: “metals such as BiTeI” There is no citation, but to the best of my knowledge, BiTeI is a semiconductor. Also, $\text{CH}_3\text{NH}_3\text{PbI}_3$ (PNAS 115, 9509, 2018), an organic-inorganic perovskite, might be relevant reference system.

- line 71: "We used a quartz lambda/4 plate" Can a single-material (= zero-order) waveplate cover the full photon energy range (2.2 eV to 3 eV) used in the measurements? Did the authors check the polarization for a larger range of wavelengths than the ones given in Fig. S2?

- line 112: "The crystal symmetry group is reduced to Cs" Reduced with respect to what?

- line 112, 113: "out of plane inversion symmetry breaking" This, or a very similar wording occurs a number of times in the text. In my book, inversion symmetry is a point symmetry that that exists or not. Maybe something like "fields along the out-of-plane direction break inversion symmetry"

- line 131: "...THz emission is not observed at the exciton excitation... This shows that the CPGE at the interband is in fact ultrafast, whereas the CPGE is not." CPGE at the interband -> CPGE triggered by interband excitation. The sentence is confusing, what is it supposed to imply? That the currents, which give rise to the CPGE, are present instantaneously (within the time resolution of the experiment, which is not specified in the manuscript) after optical excitation? Or that they decay on a femtosecond/picosecond time scale? The discussion of the THz emission dynamics is not satisfying. What mechanism for THz emission from the excitons could be expected? Radiative transitions from a higher-lying excitonic state to a lower lying one? Or non-radiative transitions from a higher-lying excitonic state to lower-lying one, generating optical phonons which then decay emitting THz radiation? If the lowest-energy exciton is optically excited, I would naively expect none of those processes, and it does not seem surprising that no THz emission is observed. In this situation, however, the absence of THz emission does not give information about the dynamics (neither generation, nor relaxation dynamics) of the excitonic CPGE, and the conclusion that "exciton CPGE is much slower" (again, in what sense? Buildup, decay, drift velocity,...?) is questionable.

- line 75: "frequency, f " -> frequency f

- line 76: "using a phase-sensitive technique" The lock-in-phase is never discussed. Is it stable when the photon energy is changed?

- line 82: " $S = 1/2$ " -> $J = S = 1/2$

- line 90: "group velocity has opposite polarity" -> group velocity has different sign, orientation

- line 171: "3.06 eV using an amplified Ti-sapphire laser at 810 nm" -> 3.06 eV using a frequency-doubled Ti-sapphire laser operating at a center wavelength of 810 nm

- line 203: "interband" -> interband transition

- line 211: "measured at zero bias, at steady state there is still" -> measured at zero external bias, at there is still [it seems to me that "steady state" and "zero external bias" mean the same thing here, but "zero bias" is much more clear]

- line 217, 218: The authors state that the small built-in field is sufficient to dissociate the exciton. They also write that the exciton binding energy is > 200 meV. What is the electric field within the exciton, that the built-in field would have to approach to dissociate it?

- line 227: $R = [(n-1)^2 - (\kappa-1)^2]/[(n+1)^2 - (\kappa+1)^2]$ at which angle? I can only memorize the textbook equation $R = [(n-1)^2 - \kappa^2]/[(n+1)^2 - \kappa^2]$ for normal incidence.

- line 228: the wording “dependent of either n , or κ , or both (via KK relation)” is confusing, as n and κ are inherently related to one another by the Kramers-Kronig (the abbreviation should be introduced) relations

- line 241: “circularly polarized reflectivity of the exciton band in 2D PEPI” Does this mean that the excitons within the $n = 1$ ($j \neq 0$) manifold are non-degenerate?

- Figure 3, especially C and D could use some formatting. “a.u.” \rightarrow a. u., (short) blanks before and after the “=”, replace the “to the power of zero” with a “ 0 ” (I assume that the publisher will do so in the main text of the proofs if the manuscript reaches that point, but I think it would be good to do so even earlier as any real number to the power of zero is basically a “1”). The insets in C should not overlap with the data points. It should be made clear at what time delay the data are taken.

Supplement:

- Please increase the font size in Figs. S3 and S5 C so that they can be read in a printout.

- line 194 “this” \rightarrow the observed step height

- Please switch the caption of Fig. S9 and the text paragraph starting out with “in Fig. S10”.

- line 240: $\psi = 24$ ($1 \pm 176\%$), goodness of fit 0.74. The formatting is confusing. What is the value of ψ , what the error? With respect to which test criterion is the goodness of fit 0.74 – is this good or bad?

- table S5: “all other elements do not affect the fitting results” Does this mean that no information about those parameters is derived from the data, or that they are close to zero?

Reviewer #2 (Remarks to the Author):

Referee Report on the paper “Circular photogalvanic spectroscopy of Rashba coupling in 2D hybrid organic-inorganic 2D perovskite quantum wells”, by Xiaojie Liu et al.

1. Page 6, line 227: instead of

$$R = \frac{(n-1)^2 - (\kappa-1)^2}{(n+1)^2 + (\kappa+1)^2}$$

should be

$$R = \frac{(n-1)^2 + \kappa^2}{(n+1)^2 + \kappa^2}$$

2. The term ‘multiple quantum wells’ used in the manuscript for the 2D hybrid perovskites is at odds with the term ‘composite material’ used in Conclusions in Ref. [19]. The DFT calculations of Even et al. [19] rule out the applicability of models based on an ultrathin quantum well with finite confinement barriers. Thus, the concept ‘multiple quantum wells’ for the material under consideration may be misleading.
3. As far as I understand, the time-inversion symmetry imposes the relation $R(\text{RCP}, \theta) = R(\text{LCP}, -\theta)$ and $R(\text{LCP}, \theta) = R(\text{RCP}, -\theta)$, where θ is the incidence angle and RCP, LCP mean the right and left circular polarizations. It follows then that the degrees of circular polarizations in the reflectivity spectrum at $\phi = 0$ and 180° should differ in sign only which is not the case in the experimental spectra in Fig. S14. What is the reason for this discrepancy?
4. Is it true that at $\phi = 90^\circ$ the reflectivities for RCP and LCP should coincide?
5. Abstract: “. . . the novel CPGE response of the excitons show spin dependent photocurrent originated from the resonant circular reflectivity.” It is clear that the circular-polarization sensitive reflectivity and CPGE are both allowed by the C_s point-group symmetry and evidently are related to the Rashba splitting. However this does not mean that one is originated from the other. The alternative can be the following three-stage mechanism: (i) the excitation of spin-polarized excitons, (ii) the dissociation of excitons into spin-polarized electrons and holes, (iii) the spin-galvanic effect described, e.g., in Refs. Nature **417**, 153 (2002); Phys. Rev. Lett. **119**, 256801 (2017).
6. Has the observed CPGE response common with the observation of photocurrents for the excitation of excitons in GaAs QWs at room temperature, Phys. Rev. Lett. **109**, 216601 (2012)?

The submitted paper presents results of fundamental experimental research of a very fashionable material to study these days. It can be published in Nature Communications if the revised version takes the above comments into account.

Reviewer #3 (Remarks to the Author):

The manuscript presents circular photogalvanic effect (CPGE) on the layered halide perovskite phenylethylammonium lead iodide ((PEA)₂PbI₄). As far as I know, this is the first example of CPGE performed over a layered perovskite. The results clearly show a spin photocurrent which may lead to interesting applications in spintronics. The separated discussion over the interband and exciton signals is extremely interesting and well presented. Therefore, I support publication of the manuscript with three remarks/corrections to consider:

- 1) I would like to see much more said on the structure of (PEA)PbI₄. What is the crystal group of the room temperature structure, does it present centrosymmetry? There is a remarkable lack of references concerning the material although it has been studied on several occasions in the literature.
- 2) How does the thickness of the film impact the results?
- 3) How are the numbers in Figure 1A established?

Response to Reviewer 1

The authors report the observation of a circular photogalvanic effect, and of optically driven spin currents, respectively, in a layered organic-inorganic perovskite-like Ruddlesden-Popper compound. To the best of my knowledge, it is the first time the effect is observed in this class of materials. The authors do not only give direct prove of a significant Rashba effect in this layered organic-inorganic compound, but thanks to the high quality of the data and a careful quantitative analysis they also draw conclusion about the symmetry of the crystal underlying the effect. The paper is overall well written. The results are of interest not only to the scientific community working with organic-inorganic perovskites, but more generally to researchers with an interest in spin effects in semiconductors. In this sense, the manuscript absolutely fulfills the standards of Nature Communications. However, I have several questions concerning the experimental details and the interpretation of the data, see below. If those can be fully addressed, I will recommend the manuscript for publication.

Detailed comments:

Comment (1): *The excitonic spectrum consists of two peaks. The one at ~2.4 eV is dominant in the reflectivity measurements on crystals (S13, S15) and in the transmission experiments on thin films (Fig. 1B, red curve). The one at 2.3 eV is prominent in the photocurrent measurements on single crystals (Fig. 1B, black symbols) as well as in the CPE action spectrum of the single crystals (Fig. 2B), with the high-energy feature being almost absent in these measurements. The authors conclude that the 2.4 eV feature is intrinsic to the material and assign the low-energy peak to defects. They also conclude that the CPE action spectrum is not the derivative of the absorption spectrum. I think that the comparison between the CPE spectrum of the crystal and the absorption spectrum of the film (as well as to the reflectivity data) is problematic. Crystal and thin film (surface which is also measured in reflectivity, edges of sheets, internal interfaces) could differ in their optical and electronic properties because of differences in their chemical composition, band bendings etc. The two transitions should be discussed carefully. Photoluminescence experiments on thin films and single crystals, similar to the ones in Lingling Mao's "Hybrid Dion-Jacobson 2D Lead Iodide Perovskites" seem simple and might help with the interpretation of the transition. At present, only a single photoluminescence spectrum is given (Fig. 1B). It is assigned to a crystal in the figure caption and to a thin film in the main text.*

Answer 1: We appreciate such detailed comment and good suggestion from the reviewer that may help us improve the scientific value of this work.

We answer comment 1 in three parts; (i)-(iii).

1(i): We have double-checked with the original lab record, and confirm that the PL emission spectrum is in fact from the crystal not the film. We have corrected this mistake in the main text.

1(ii): We agree with the reviewer's concern of potential problems of comparing data from film and single crystal samples. Yet, as will be detailed below, the difference in the basic optical properties of film and crystal is not substantial for PEPI because the film grows in the z-direction in any case. We do not think that presenting the absorption spectrum of a film sample will affect the results in this study, since the presented spectra in Fig. 1 are for introduction purposes only. In addition; the absorption from crystal

does not contain much information in any case... Our work is about CPGE in single crystal sample. Nevertheless we have decided to remove the absorption spectrum from Fig. 1B and place it in the S.I. section, in order to avoid unnecessary confusion.

Unfortunately we do not have the capability to measure the absorption of exfoliated crystal that has a few layers with the size of a few micrometers, as was done in the ref [Fieramosca, A. et al., ACS Photonics 2018, 5, 4179–4185]. For the purpose of rough comparison, we have plotted our data on top of the figure taken from the ref of Fieramosca et al., as shown in Fig. R1 (a). As can be seen, other than a small blue shift (~ 5 meV), the absorption of film and single crystal is essentially the same, with nearly identical absorption edge. Similar case was also reported in Fig. 1G of [Science 355, 1288–1292 (2017)].

Figure R1 (b) shows a comparison of the PL spectra from a film and crystal sample (our data). As expected, the PL of crystal is blue-shifted by about 20 meV from that of the film, and the film PL spectrum shows more contribution at lower energy, possibly from excess defects at grain boundaries. This trend is also consistent with Fig. 1G in [Science 355, 1288–1292 (2017)].

Fig. R1. (a) Comparison of absorption and PL spectra with those of ref [Fieramosca, A. et al., ACS Photonics 2018, 5, 4179–4185]. The solid lines are for crystal from that reference, whereas the dashed lines are our data (the PL is from crystal and the absorption is from a film). (b) Comparison of the PL spectra for a thin film sample (black) and a single crystal (red) from our arsenal.

1(iii): While the exciton higher energy peak at 2.4 eV is unambiguously assigned to the $1s$ exciton transition, it is also true that the nature of the lower energy peak at 2.3 eV is less clear. The reason that the 2.3eV peak is more prominent in the photoconductivity (Fig. 1B, black) action spectrum is that the PC related to defects in the sample *lives longer*, and thus is apparently stronger in the measured steady state conditions. This is a well-known phenomenon in disordered films, see for example *Phys. Rev. Lett.* 101, 037401 (2008).

We have conducted the suggested PL measurements and the results are now shown as a new Fig. 1B.

Comment (2): *While the CPE experiments and their symmetry analysis give direct evidence of the Rashba splitting and information about its directionality, the interpretation of the Terahertz experiments seems less straightforward to me. What is the origin of the THz emission? That “THz emission is proportional to the time derivative of the photocurrent” does not seem obvious to me. Are the authors referring to the Bremsstrahlung emitted as the electrons relax from the initially optically excited state (with finite group velocity) to the band minimum (with*

group velocity zero)? Why can the signal not result from optical phonons generated during the relaxation of carriers to the band edge? This would naturally explain that for the excitonic transition no THz emission is observed, as the excess energy is insufficient for optical phonon generation. What is the origin of the ~1 THz periodic modulation of the signal? How are those periodic modulations consistent with the interpretation of the signal as a change in current? How are the data points in Fig. 3C, 3D extracted from the data in 3B – at what time delay?

Answer 2: We appreciate reviewer’s comment. Again we would like to address this comment in few parts:

Comment 2(i): What is the origin of the THz emission?

Answer 2(i): The THz emission process can be described using Maxwell equations, where the inhomogeneous wave equation in the presence of electric charges and current has the form:

$$\nabla^2 E(r, t) - \frac{1}{c^2} \frac{\partial^2}{\partial t^2} E(r, t) = -\mu_0 \frac{\partial}{\partial t} \left(j + \frac{\partial P}{\partial t} \right)$$

Thus, time varying current and second time derivative of material polarization act as sources for electromagnetic radiation. However, only ultrafast current results in pulsed emission of terahertz radiation that corresponds to sub-picosecond electric field. These ultrafast currents are behind terahertz emission using ‘Austin Switches’ that are commercial photoconductive antennas. These switches are used as terahertz sources based on low-temperature grown GaAs and InAs, where external bias provides the needed carriers acceleration to initiate current flow. The plot below shows ultrafast flow of photocurrent upon photo-excitation, of which derivative is in fact the terahertz emission signal.

Fig. R2: The transient response of a THz emitter initiated by an ultrafast pulse (blue line). The black line is the transient photocurrent, whereas the red line is the THz emission, which is in fact, the time derivative of the photocurrent transient response; in agreement with the theory.

The experiments for circular photo-galvanic effect were done with femtosecond pulse excitation. As the reviewer correctly pointed out, the photogenerated current flows with finite group velocity when the photo-excited carriers with 3.1 eV pump relax to conduction band minima (with zero group velocity), before recombination sets in, which is a relatively slow process. Since the pulse excitation is 45-56 fs in duration, the process of photo galvanic current is estimated to happen in the femtosecond time scale

associated with momentum scattering time of the material. We note that terahertz emission associated with fast photo galvanic currents has been presented in studies referenced below:

1. Obraztsov, P.A., Lyashenko, D., Chizhov, P.A., Konishi, K., Nemoto, N., Kuwata-Gonokami, M., Welch, E., Obraztsov, A.N. and Zakhidov, A., 2018. Ultrafast zero-bias photocurrent and terahertz emission in hybrid perovskites. *Communications Physics*, 1(1), p.14.
2. Kinoshita, Y., Kida, N., Miyamoto, T., Kanou, M., Sasagawa, T. and Okamoto, H., 2018. Terahertz radiation by subpicosecond spin-polarized photocurrent originating from Dirac electrons in a Rashba-type polar semiconductor. *Physical Review B*, 97(16), p.161104.
3. Sirica, N., Tobey, R.I., Zhao, L.X., Chen, G.F., Xu, B., Yang, R., Shen, B., Yarotski, D.A., Bowlan, P., Trugman, S.A. and Zhu, J.X., 2019. Tracking ultrafast photocurrents in the Weyl semimetal TaAs using THz emission spectroscopy. *Physical review letters*, 122(19), p.197401.

Comment 2(ii): Why can the signal not result from optical phonons generated during the relaxation of carriers to the band edge? What is the origin of the ~1 THz periodic modulation of the signal? How are those periodic modulations consistent with the interpretation of the signal as a change in current?

Answer 2(ii): The process of terahertz emission from optical phonons is unlikely to be the reason behind our observation, as such process would have a different characteristic transient; namely long-lived periodically modulated signal rather than a single cycle emission as measured. As the reviewer correctly pointed out, the n=1 two dimensional hybrid perovskite has optical phonons in this frequency range. In fact, we measured the optical phonon modes using terahertz transmission through the crystal w.r.t. to reference substrate. The observed phonons can be observed at 0.78 THz and 1.6 THz, respectively; probably associated with Pb-I-Pb rocking vibration and Pb-I stretching bonds (Ref: M. Sender et al. *Materials Horizons* **3**, 6, 613-620 (2016)). The reviewer is also correct in pointing out that the terahertz emission has emission primarily at 1 THz while the frequency ranges at 0.8 THz and above 1.6 THz is suppressed. This is due to the fact that we measured terahertz emission in *transmission mode* through the crystal, where photo-absorption and emission happen from limited thickness of the crystal. The transmission of the emitted radiation through the thickness of 7-10 μm of the crystal would cause subsequent absorption of the signal to a significant amount at resonance with those phonon frequencies. Therefore the measured THz radiation is peaked at 1 THz where the dip between the two phonons is observed.

We thus conclude that the presence of emitted signal at complementary frequencies to those of the phonons indicates process other than of phonon emission.

Moreover, the helicity dependence of the optical excitation points to spin-dependent process which is present in CW measurements, rather than simply due to phonons. Terahertz emission due to photo-galvanic currents provides additional information, where the photocurrents are ultrafast during the process of relaxation from inter-band state to conduction band minima; as they are expected to be. It is noteworthy that from circular photo-galvanic effect we expect the current direction to change when excited with opposite helicity of the circular polarization pump (as presented in the references above). However, similar to the CW measurements, the PEPI crystal has other THz emission processes (not necessarily the same mechanisms as in the CW measurements), in addition to the CPGE effect, namely

the linear polarization dependent transient photocurrent (L), and polarization independent transient photocurrent (D). Therefore similar to the CW measurements, we see a modulation of the THz emission related to the CPGE effect, on top of a background THz signal.

The plots below show the transient THz emission and its Fourier transform spectrum. For comparison, the plot to the right shows the measured absorption spectrum in the THz range through the PEPI crystal that shows the optical phonon modes.

Fig. R3: (A) The transient THz emission and (B) its Fourier transform spectrum measured from PEPI crystal, compared to (C) the crystal absorption spectrum measured in the THz range. The spectrum in (C) is decomposed into two phonon modes as indicated.

Comment 2(iii): How are the data points in Fig. 3C, 3D extracted from the data in 3B – at what time delay?

Answer 2(iii): The data points for the E-Field in Figs. 3C and 3D are extracted as ‘peak-to-peak’ values in the terahertz emission signal. The peak-to-peak value of the E-field corresponds to the amplitude of the photogalvanic current, J , as defined in the phenomenological model described in equation (1) in the text:

$$J_{x'}(\alpha) = C1\sin(2\alpha) + C2 \cos(2\alpha) + L1\sin(4\alpha) + L2\cos(4\alpha) + D,$$

For clarification, we have relabeled the Y-axis of the plots in Fig. 3 to read ‘peak-to-peak E Field’. The definition of peak-to-peak has been marked in schematic in Fig. 3a, which seems to have been cut off in the submitted file. We have updated that and marked the points in Fig. 3b as asked. Please find below the updated Fig. 3 and its caption. Corresponding changes have been made in the manuscript. The definition is consistent with the previous studies and references mentioned above. We have also added the above references to the manuscript, in order to provide the readers other relevant terahertz emission measurements related with the photo-galvanic effects in other materials.

Revised Fig 3 (in the text). Terahertz (THz) emission due to ultrafast photogalvanic (PGE) currents in 2D-PEPI crystal: A, schematic illustration of the experimental set-up for the THz emission measurements. The angles α , θ , and ϕ are denoted. B, THz time domain emission waveforms measured at $\lambda/4$ wave plate angle, α , of 0° , 45° and 135° as denoted, that correspond to linear polarized (LP), right circularly polarized (RCP) and left circularly polarized (LCP) light. The two dashed black lines mark the times where positive and negative peaks of the terahertz emission were determined. The peak-to-peak values are determined from the addition of the absolute positive and negative values as marked by the dots. The inset shows a null signal obtained when resonantly excited at the exciton band (2.34 eV). C. Terahertz field emission vs. the rotation angle, α , photogenerated using 3.06 eV pulsed excitation at $\theta=45^\circ$. The red line through the black data points is a fit using Eq. (1). The inset shows the obtained relative values of the coefficients C1, L2 and D; D. Similar measurements as in panel C, for incident angle, $\theta=0^\circ$. The fit using Eq. (1) shows majority contribution from L2. The blue dashed line marks the noise level corresponding to zero emission field.

Comment 3: *“...THz emission is not observed at the exciton excitation... This shows that the CPGE at the interband is in fact ultrafast, whereas the CPGE is not.” CPGE at the interband -> CPGE triggered by interband excitation. The sentence is confusing, what is it supposed to imply? That the currents, which give rise to the CPGE, are present instantaneously (within the time resolution of the experiment, which is not specified in the manuscript) after optical excitation? Or that they decay on a femtosecond/picosecond time scale? The discussion of the THz emission dynamics is not satisfying. What mechanism for THz emission from the excitons could be expected? Radiative transitions from a higher-lying excitonic state to a lower lying one? Or non-radiative transitions from a higher-lying excitonic state to lower-lying one, generating optical phonons which then decay emitting THz radiation? If the lowest-energy exciton is optically excited, I would naively expect none of those processes, and it does not seem surprising that no THz emission is observed. In this situation, however, the absence of THz emission does not give information about the dynamics (neither generation, nor relaxation dynamics) of the excitonic CPGE, and the conclusion that “exciton CPGE is much slower” (again, in what sense? Buildup, decay, drift velocity,...?) is questionable.*

Answer 3: We have clarified the sentence that the referee pointed out. The sentence now reads “THz emission is not observed when the samples are excited at the exciton absorption band at 2.4 eV with similar photon density (4×10^{17} photons/cm³) . This shows that the CPGE at the conduction band for interband transition at 3.1 eV is in fact ultrafast, whereas the CPGE observed at the exciton band in the CW measurement is less so. This indicates that the exciton-related photocurrent decays slower than sub-picosecond timescale, and therefore it does not generate terahertz emission.”

The reviewer makes a relevant point that terahertz emission of sub-picosecond pulses only originates from femtosecond/picosecond photocurrent decay. Photocarriers that recombine at a slower rate cannot give THz emission and hence, is outside the time-interval of our technique. We note that the presence of terahertz emission proves that the associated photocurrent relaxes fast, as expected. The lack of terahertz emission related to exciton indicates that the photocurrent generated from exciton dissociation might be slower than that excited at the interband, but still have spin polarization. Such a mechanism may be related to spin-PGE, where the exciton dissociation with circular polarized light generate spin aligned photocarriers upon dissociation, which decay in time by spin relaxation processes. As added to the manuscript, the spin relaxation time in PEPI at room temperature is of the order of few ps and thus cannot generate THz radiation.

Second part of the reviewer’s comment enquires about the expected response of excitons in the terahertz spectral range. We note that since excitons are bound-electron-hole pairs, the photocurrent is generated *after* exciton dissociation into free electron and hole. The binding energy in 2D-PEPI is in range of 200 meV. Thus the dissociation of excitons in the absence of external bias may result from electric field provided by traps, defects or surface fields.

Comment (4): *The quality of the writing in the abstract needs to be improved to match the one in the main text: continuum bands -> bands (are continuous by definition), existence of Rashba splitting of 35 meV -> existence of a Rashba splitting of 35 meV, CPE at interband -> CPE associated with interband transitions, spin-polarized hot photocarriers whose spins are split in momentum space -> ... Also, on “2D-PEPI is an*

exemplary hybrid organic-inorganic halide perovskite semiconductor“: In the main text, the perovskite and the Ruddlesdon-Popper structures are properly distinguished. This might also make sense here.

Answer 4: We thank the reviewer’s attention to these details, and the rephrasing suggestions. We have re-written the abstract accordingly.

Comment 5: Line 40: „Pbl layers with n the number of inorganic Pbl layers“, line 43: “inorganic Pbl well and organic spacer wall”: in general, the quantum wells are not inorganic, but formed by a hybrid organic-inorganic structure, as the authors state correctly in the first sentence of the paragraph. The wording “inorganic Pbl layers” is a little confusing, since the reader expect (from abstract and introduction) quantum wells with a perovskite-like structure, rather than a metal halogenide salt. Please use more specific wording.

Answer 5: We appreciate the meticulous and attentive manner of the reviewer and have made revisions accordingly.

Comment 6: I believe the authors use the term “hot electrons” for the electronic state created by the optical excitation. In the literature, in contrast, the term is used for an electronic system that has reached quasi-equilibrium (i. e., electrons have relaxed to the band edges) and can be described by an electronic temperature (therefore the term “hot”) which is different from (and typically much higher than) the one of the phonon bath. I suggest that the authors discriminate clearly between the optically excited state and the electronic system in quasi-equilibrium. This relates to line 94, where the authors state that the hot carrier relaxation/thermalization time determines the decay of the CPGE current. I would rather think that it is the momentum scattering and randomization time which dominates the CPGE decay, in line with a publication by one of the authors (Appl. Phys. Lett. 109, 193903 (2016)).

Answer 6: We accept the suggestion and have made changes accordingly.

(7) lines 42/43: multiple quantum wells structure -> multiple quantum well structures

A6: Change has been made.

(8) line 49: a/the spin-related photocurrent

A8: Change has been made.

(9) line 56/57: “PL spectra of PEPI thin film” inconsistent with figure caption that claims that PL spectrum of a crystal is shown. Also, there is only one spectrum in the figure.

A9. Change has been made. Following another reviewer’s suggestion, we have replaced Fig. 1B with a new one, and presented PL in a new figure in SI (Fig. S8

(10) line 59/60: “PL spectrum has a dominant band at 2.35 eV that is associated with a large exciton binding energy > 200 meV” band -> peak; the peak is associated with the exciton (not in any obvious way with its binding energy)

A10: We appreciate the reviewer’s comment and have made changes accordingly. Again, the description of PL spectrum is now in SI, the new Fig.S8.

(11) line 67: “metals such as BiTeI” There is no citation, but to the best of my knowledge, BiTeI is a semiconductor. Also, CH₃NH₃PbI₃ (PNAS 115, 9509, 2018), an organic-inorganic perovskite, might be relevant reference system.

A11: Indeed BiTeI is a polar semiconductor. We have made the correction and added the reference of MAPbI as suggested.

(12) line 71: “We used a quartz lambda/4 plate” Can a single-material (= zero-order) waveplate cover the full photon energy range (2.2 eV to 3 eV) used in the measurements? Did the authors check the polarization for a larger range of wavelengths than the ones given in Fig. S2?

A12: The quartz lambda/4 plate is from Thorlab, model AQWP05M-600 - Ø1/2" Mounted Achromatic Quarter-Wave Plate, Ø1" Mount, 400 - 800 nm. It covers the energy range from 1.55-3.1eV. Yes, we have measured as low as 640nm (1.93eV) using a diode laser as excitation source; we found that at 640nm there is null for both CPGE and the background DC current. This is no surprise since there is no absorption at such low energy (absorption edge is at 2.3 eV).

(13) line 112: “The crystal symmetry group is reduced to Cs” Reduced with respect to what?

A13: Upon similar comments from another reviewer, we have referred to a few new references on the crystal structure of 2D-PEPI, and have adopted the most up-to-date identification, which is triclinic structure (space group $P\bar{1}$). It is nevertheless centrosymmetric. We have revised the description accordingly.

(14) line 112, 113: “out of plane inversion symmetry breaking” This, or a very similar wording occurs a number of times in the text. In my book, inversion symmetry is a point symmetry that exists or not. Maybe something like “fields along the out-of-plane direction break inversion symmetry”.

A14: Thanks for the suggestion of a more precise phrase. We have made corrections accordingly.

(15) line 131: “...THz emission is not observed at the exciton excitation... This shows that the CPGE at the interband is in fact ultrafast, whereas the CPGE is not.” CPGE at the interband -> CPGE triggered by interband excitation. The sentence is confusing, what is it supposed to imply? That the currents, which give rise to the CPGE, are present instantaneously (within the time resolution of the experiment, which is not specified in the manuscript) after optical excitation? Or that they decay on a femtosecond/picosecond time scale? The discussion of the THz emission dynamics is not satisfying. What mechanism for THz emission from the excitons could be expected? Radiative transitions from a higher-lying excitonic state to a lower lying one? Or non-radiative transitions from a higher-lying excitonic state to lower-lying one, generating optical phonons which then decay emitting THz radiation? If the lowest-energy exciton is optically excited, I would naively expect none of those processes, and it does not seem surprising that no THz emission is observed. In this situation, however, the absence of THz emission does not give information about the dynamics (neither generation, nor relaxation dynamics) of the excitonic CPGE, and the conclusion that “exciton CPGE is much slower” (again, in what sense? Buildup, decay, drift velocity,...?) is questionable.

A15: We believe that we have responded to this comment in detail in our responses A2 and A3 above

(16) line 75: “frequency, f ” -> frequency f

A16: We have made the suggested change.

(17) line 76: “using a phase-sensitive technique” The lock-in-phase is never discussed. Is it stable when the photon energy is changed?

A17 : The lockin amplifier phase does not change with the excitation photon energy, since the modulation frequency is much ‘slower’ than the CPGE process at all excitation photon energies used here.

(18) line 82: “ $S = 1/2$ ” -> $J = S = \frac{1}{2}$

A18: OK; we have corrected accordingly.

(19) line 90: “group velocity has opposite polarity” -> group velocity has different sign, orientation.

A19: We have changed the wording as suggested.

(20) line 171: “3.06 eV using an amplified Ti-sapphire laser at 810 nm” -> 3.06 eV using a frequency-doubled Ti-sapphire laser operating at a center wavelength of 810 nm

A20: We have made the suggested change.

(21) line 203: “interband” -> interband transition

A21: Done

(22) line 211: “measured at zero bias, at steady state there is still” -> measured at zero external bias, at there is still [it seems to me that “steady state” and “zero external bias” mean the same thing here, but “zero bias” is much more clear]

A22: We actually have to include both terms since they describe different conditions. ‘Zero bias’ means externally applied voltage, whereas ‘steady state’ describes the excitation light source (from CW laser or Xenon lamp), when the measurements are done by modulating the light source intensity.

(23) line 217, 218: The authors state that the small built-in field is sufficient to dissociate the exciton. They also write that the exciton binding energy is > 200 meV. What is the electric field within the exciton, that the built-in field would have to approach to dissociate it?

A23: We gave a detailed description about the internal electric field in the device used for PGE measurements in the SI (MM4). The internal electric field in the device is possibly due to photothermal (Seebeck effect) and/or from a small asymmetry of the two gold electrodes work functions. Based on the comparison between PC @ (-5V) current and the DC offset current in PGE (D in equ. (1)), we estimate the internal electric field E_{PGE} at zero bias condition to be 500 V/cm . This is insufficient to ionize the excitons in PEPI, but is needed for the photocurrent AFTER the exciton dissociation. We have changed our previous statement based on this more detailed estimate. We now claim that the excitons in PEPI dissociate in edge states and defects as previously discussed in the literature.

(24) line 227: $R = \frac{[(n-1)^2 - (\kappa-1)^2]}{[(n+1)^2 - (\kappa+1)^2]}$ at which angle? I can only memorize the textbook equation $R = \frac{[(n-1)^2 - \kappa^2]}{[(n+1)^2 + \kappa^2]}$ for normal incidence.

A24: The reviewer is correct. We have discarded that formula and used a more relevant equation which includes the incident angle.

(25) line 228: *the wording “dependent of either n , or κ , or both (via KK relation)” is confusing, as n and κ are inherently related to one another by the Kramers-Kronig (the abbreviation should be introduced) relations*

A25: Done.

(26) line 241: *“circularly polarized reflectivity of the exciton band in 2D PEPI” Does this mean that the excitons within the $n = 1$ ($j \neq 0$) manifold are non-degenerate?*

A26: Yes; the referee is correct. The exciton in semiconductors involve electron and hole and thus has four states for $j=1/2$ electron and $j=s=1/2$ for the hole. The split of the four states is known in the literature as “Fine Structure” of the exciton. In the Tetragonal phase, for example these are split into one singlet; one z-polarized; and circular polarized pair states. These states are further split by the crystal field in structures of lower symmetry.

(27) Figure 3, especially C and D could use some formatting. “a.u.” -> a. u., (short) blanks before and after the “=”, replace the “to the power of zero” with a “ 0 ” (I assume that the publisher will do so in the main text of the proofs if the manuscript reaches that point, but I think it would be good to do so even earlier as any real number to the power of zero is basically a “1”). The insets in C should not overlap with the data points. It should be made clear at what time delay the data are taken.

A27: We have made changes as suggested.

Supplement:

(S1) Please increase the font size in Figs. S3 and S5 C so that they can be read in a printout.

SA1: We have made the change.

(S2) line 194 “this” -> the observed step height

SA2: we have made the change.

(S3) Please switch the caption of Fig. S9 and the text paragraph starting out with “in Fig. S10”.

SA3: Done

(S4) line 240: $\psi = 24$ (1+-176%), goodness of fit 0.74. The formatting is confusing. What is the value of ψ , what the error? With respect to which test criterion is the goodness of fit 0.74 – is this good or bad?

SA4: We appreciate the careful reading from the reviewer. Firstly, ψ (ψ) is the angle between the current direction in the measurement (determined by the electrodes orientation) and the b-axis of the crystal (see Fig. 1C). Its value is 24° , with uncertainty of 42° . This angle is unknown at the time of device fabrication, due to the small size of the crystal (<5mm, as shown in Fig. S5), and its irregular shape (usually elongated, see Fig. S1A), we chose the best part for each device, and therefore ψ (ψ) varies from device to device. Secondly, we have demonstrated in SI-MM9 that there is a way to find this angle based on THz emission field strengths along two orthogonal directions (see Fig. S8), that is equ. (S13).

Since the inversion symmetry break within the $[\text{PbI}_6]^{4-}$ plane is small, at normal incidence, CPGE effect is expected to be small, that was why the error bar was so big. The goodness of fitting refers to the adjusted R-squared in the least-square fitting used. A goodness of 0.74 is considered as okay (not very good). We agree that the presentation format was confusing and have changed it to the expression format used in Table S2.

Table S5: “all other elements do not affect the fitting results” Does this mean that no information about those parameters is derived from the data, or that they are close to zero?

SA5: Fitting parameters used in Fig. 2C are 8 non-zero components of χ & γ tensors including: 2 non-zero components of χ tensor (χ_{xxz} and χ_{yyz}), and 2 non-zero components of γ tensor (γ_{xy} and γ_{yx}) for out-of-plane inversion asymmetry along z-axis; 3 non-zero components of χ tensor (χ_{xxy} , χ_{yxx} and χ_{yyy}), and 1 non-zero components of γ tensor (γ_{xz}) for in-plane inversion asymmetry along y-axis, and angle ψ between the current direction and the crystal a-axis as defined in Fig. 1C. Among all 9 parameters, the ones that matter are described in SI-MM9 and listed in SI, Table S5. ‘all other elements do not affect the fitting results’ means that these parameters do not affect the fitting result in noticeable way, and their values are not important. We have added this additional information to SI-MM11 to provide a clearer view for the readers.

Response to Reviewer 2

Comment 1: Page 6, line 227: instead of

$$R = \frac{(n - 1)^2 - (\kappa - 1)^2}{(n + 1)^2 + (\kappa + 1)^2}$$

should be

$$R = \frac{(n - 1)^2 + \kappa^2}{(n + 1)^2 + \kappa^2}$$

Answer 1: We appreciate the reviewer's correction of the erroneous formula in our previous submission. Based on the new experimental results and the reviewer's suggestion, we have now considered circular reflectivity NOT the mechanism for CPGE current upon exciton excitation. Consequently, all discussions about that topic have been removed from the main text.

Comment 2: The term 'multiple quantum wells' used in the manuscript for the 2D hybrid perovskites is at odds with the term 'composite material' used in Conclusions in Ref. [19]. The DFT calculations of Even et al. [19] rule out the applicability of models based on an ultrathin quantum well with finite confinement barriers. Thus, the concept 'multiple quantum wells' for the material under consideration may be misleading.

Answer 2: The term multiple quantum wells is a description of systems in which the HOMO/LUMO gap changes in space to form potential wells (more specifically, wells that are so confined spatially that quantization effects become important). This is certainly the case for this material. That is why most researchers in the literature refer to 2D perovskites as multiple quantum wells, see, for example, the recent paper by Even, J. et al. [NATURE COMMUNICATIONS | (2018) 9:2254, 'Scaling law for excitons in 2D perovskite quantum Wells'].

We consider this inconsistency more of an issue of semantics, and have made necessary revision to clarify it, both in the main text and SI.

Comment 3: As far as I understand, the time-inversion symmetry imposes the relation $R(\text{RCP}, \theta) = R(\text{LCP}, -\theta)$ and $R(\text{LCP}, \theta) = R(\text{RCP}, -\theta)$, where θ is the incidence angle and RCP, LCP mean the right and left circular polarizations. It follows then that the degrees of circular polarizations in the reflectivity spectrum at $\varphi = 0$ and 180° should differ in sign only which is not the case in the experimental spectra in Fig. S14. What is the reason for this discrepancy?

Answer 3:

The relation given by the referee only applies for materials which are fully reflective ($R = 1$), which is not the case for the experiments. Indeed, in this limit, the relation given by the referee proves that there is *no* dichroism at normal incidence. However, for partially reflective materials, the time reversal operation does not correspond to $\theta \rightarrow -\theta$, $(\pm) \rightarrow (\mp)$ (where \pm indicates sign of light polarization. This is demonstrated below:

Fig. R1. Schematic drawing of light pathways upon oblique incidence at the interface of a partial reflective material. It shows the *inequivalence* of incident angle reversal and time reversal.

The blue arrows indicate the Poynting vector, the yellow arrows indicate the photon angular momentum. The above shows that time reversal changes the boundary conditions of the scattering problem, resulting in a configuration different than the experimental one.

The operation that connects θ to $-\theta$ scattering conditions is a rotation about the normal direction of 180° . In this case, a symmetry analysis proceeds as shown below:

Fig. R2. Schematic drawing of light pathways upon oblique incidence in the two scenarios of scattering and time reversal operation. This shows the *equivalence* of incident angle reversal with the two-fold rotation about the normal direction of 180° .

The red arrows indicate \hat{n} , which is defined as the axis along which dichroism occurs, and is determined by structural properties of the material. Upon 180° rotation about surface normal, the out-of-plane component of \hat{n} is invariant, while the in-plane component switches sign. The operation is therefore a symmetry of the system only if \hat{n} is entirely out-of-plane, in which case we can make the following inference: $R(\theta, \pm) = R(-\theta, \pm)$, indicating that DCP

would be the same for θ and $-\theta$. The strong violation of this relation observed experimentally reveals that \hat{n} must have an in-plane component (more specifically, an in-plane component with nonzero overlap with incoming Poynting vector).

Comment 4: Is it true that at $\phi = 90^\circ$ the reflectivities for RCP and LCP should coincide?

Answer 4: The reviewer is correct. LCP (σ_-) and RCP(σ_+) are similar other than their magnitude, which is also explained by the calculation on p16 in the SI. Fig. R3 below shows the DCP at $\phi=90^\circ$.

Fig. R3. The reflectivity spectra of 2D-PEPI measured with LCP and RCP polarizations (red and black points) at incident angle $\theta=20^\circ$ and azimuthal angle $\phi=90^\circ$. The blue line is the degree of circular polarization (DCP) obtained at this experimental condition.

Comment 5: Abstract: “. . . the novel CPGE response of the excitons show spin dependent photocurrent originated from the resonant circular reflectivity.” It is clear that the circular-polarization sensitive reflectivity and CPGE are both allowed by the Cs point-group symmetry and evidently are related to the Rashba splitting. However this does not mean that one is originated from the other. The alternative can be the following three-stage mechanism: (i) the excitation of spin-polarized excitons, (ii) the dissociation of excitons into spin-polarized electrons and holes, (iii) the spin-galvanic effect described, e.g., in Refs. Nature 417, 153 (2002); Phys. Rev. Let. 119, 256801 (2017).

Answer 5: We are grateful to the reviewer for suggesting this alternative mechanism to explain the CPGE associated with the exciton generation; and we adopted it wholeheartedly. For confirming that this mechanism may explain our results, we performed a circularly polarized transient pump-probe transmission of a PEPI crystal at room temperature. In this experiment, which is described in the revised SI in detail, we modulate the relative circular polarization of the pump and probe beam, for measuring the spin relaxation time of the exciton with superior resolution. As seen in Fig. R4 below, the spin relaxation process proceeds with two time constants (TC); a fast TC of 2.3 ps and a slower TC of 14.6 ps. The fast TC may be related to the exciton thermalization, whereas the slower TC is for thermalized excitons. As seen the spin relaxation is much slower compared to the momentum relaxation time (few tens of fs), and this explain the lack of THz emission from the photocurrent associated with the exciton SGE.

In the new version, we have changed the explanation of the CPGE at the exciton as due to SGE. We thank again the referee for this suggestion.

Fig. R4: Transient circularly polarized pump-probe transmission of PEPI thin crystal measured at room temperature using pump at 410 nm and probe at 537 nm, in resonance with the exciton band. The line through the data points is a fit using a double exponential function with time constants of 2.3 ps and 14 ps, respectively, from which we obtain an average spin relaxation time of ~ 4.5 ps. This figure has been added to the SI section.

Comment 6: Has the observed CPGE response common with the observation of photocurrents for the excitation of excitons in GaAs QWs at room temperature, Phys. Rev. Lett. **109**, 216601 (2012)?

Answer 6: We thank the reviewer for referring to an excellent paper. We believe the observation of a new type of shift-current in that work is different than the CPGE associated with excitons that we observed here. Firstly, the experimental conditions are very different: the excitation used in that reference was from two orthogonal pulse laser sources, whereas ours (Fig. 2B) was from a cw light source (Xenon lamp). The reference paper clearly stated that the new type of shift-current ‘vanishes in the continuous-wave limit’. Secondly, the crystal symmetry in that reference is C_{2v} (reduced to C_s along [110] growth), yet ours is C_s . Therefore, one expects different tensors for the different crystal groups. Finally, but most importantly, the excitation in that paper was intra-band transitions rather than the interband transition that we have studied here.

Comment 7: The submitted paper presents results of fundamental experimental research of a very fashionable material to study these days. It can be published in Nature Communications if the revised version takes the above comments into account.

Answer 7: We thank the referee for the positive recommendation and hope that our rebuttal arguments are satisfactory.

Response to Reviewer 3

The manuscript presents circular photogalvanic effect (CPGE) on the layered halide perovskite phenylethylammonium lead iodide ((PEA)2PbI4). As far as I know, this is the first example of CPGE performed over a layered perovskite. The results clearly show a spin photocurrent which may lead to interesting applications in spintronics. The separated discussion over the interband and exciton signals is extremely interesting and well presented. Therefore, I support publication of the manuscript with three remarks/corrections to consider:

Comment (1): *I would like to see much more said on the structure of (PEA)PbI4. What is the crystal group of the room temperature structure, does it present centrosymmetry? There is a remarkable lack of references concerning the materials although it has been studied on several occasions in the literature.*

Answer 1: We thank the referee for the suggestion. Indeed there have been quite a few thorough studies on the crystal symmetry group of 2D-PEPI. The first such study was ref [20] by Calabrese et al. in 1991, and we adopted the lattice parameters from their report. They concluded that the space group is $C2/m$, and the Bravais lattice is monoclinic at low

temperature (< 203K). A more recent report by Fang, H et al. (ref.[s8] (*Adv. Funct. Mater.* **28**, 1800305-1800316 (2018)) mentioned in the SI, confirmed that the space group is $P2_{1/c}$, and the crystal structure is centrosymmetric at low temperature, although the lattice constants are smaller than the previously reported values. Two independent reports, however have identified that the crystal structure at room temperature is triclinic with space group $P\bar{1}$ and centrosymmetric [Du, K. et al., *Inorg. Chem.* 2017, 56, 9291–9302; Fieramosca, A. et al. *ACS Photonics* 5, 4179–4185 (2018)] and [Chem. Mater. 2018, 30, 8538–8545]. These references show that, although there is still disagreement regarding the symmetry and crystal structure of 2D-PEPI, there seems to be a consensus that, at room temperature, the space group is $P\bar{1}$ which is centrosymmetric. For our purpose, inversion symmetry is the key. We therefore removed the details of the monoclinic crystal structure from Fig. 1A caption and the main text. In any case, in the revised version we have added a number of references that deal with PEPI crystal structure.

Comment (2): *How does the thickness of the film impact the results?*

Answer 2: All of the devices in this work were made from free standing single crystal flakes with thickness between 8-13 μm (see Fig. s5 in SI). In PEPI, such thickness is much larger than the light penetration depth (tens of nm) at all wavelengths, since the absorption coefficient is $\sim 10^5 \text{ cm}^{-1}$, ref : ACS Energy Lett. 2018, 3, 2273–2279; 2)Nanoscale, 2018, 10, 8677–8688]. Therefore we do not think that the single crystal thickness would have notable impact on the observation and interpretation of the photogalvanic effect in our samples.

Comment (3): *How are the numbers in Figure 1A established?*

Answer 3: The quantum well schematic drawing was inspired by Ref [19], Fig. 8a. The value of the organic barrier ‘wall’ was taken from ref. 19. The value for the inorganic well value was taken from the 1s exciton transition energy of 2.40 eV [Fig. 1B]. This value should have been the bulk MAPbI₃ bandgap (taken as 1.7eV) according to ref [K. Tanaka, T. Kondo, *Sci. Technol. Adv. Mater.* 2003, 4, 599. Now as the new reference [22] in the new main text]. However, upon carefully reading ref [19] and ref [22] therein, we would like to use both values from ref [22] (Tanaka).

We have made the appropriate changes in the revised manuscript.

REVIEWERS' COMMENTS:

Reviewer #1 (Remarks to the Author):

The authors fully addressed all my comments from the first round of research. I recommend to publish the manuscript in Nature Communications.

Minor corrections:

Ruddleston-Popper → Ruddlesden-Popper

wire grip polarizer → wire grid polarizer

data-points → data points

the slow process such as photothermal effect dominates → slow processes such as the photothermal effect dominate

slight asymmetry among the two Au electrodes → slight asymmetry between the two Au electrodes

due to spin-galvanic effect rather than CPGE → due to the spin-galvanic effect rather than the CPGE

same polarization than → same polarization as

upon absorption by a circular polarized pump → after creation by a circularly polarized pump

Similar situation occurs → A similar situation occurs

from ferromagnet electrode → from a ferromagnetic electrode

Rashba-Edlestein → Rashba-Edelstein

Reviewer #2 (Remarks to the Author):

My opinion is that the manuscript can be published in Nature Communications in the revised form.

In the revised version, the circular reflectivity topic has been removed from the main text. This means that my comment 3 is not related to the revised version. Nevertheless, I would like to state that the authors' answer to the comment 3 is unsatisfactory. Figure R1 in the rebuttal can in fact be applied for a process where the incident particle transforms into two particles. In the reflectivity, one has a transformation of an initial state of a particle (photon) into a final state of a particle (photon). The reflection is an extreme manifestation of the light scattering. The time invariance implies coincidence of the probabilities of the scattering processes $\alpha \rightarrow \beta$ and $T\beta \rightarrow T\alpha$, where α and β are the states participating in the scattering and T is the time reversal which, in addition to changing initial and final states, flips the wave vectors and circular polarization, so that $R(\text{RCP}, \theta) = R(\text{LCP}, -\theta)$, as is written in my previous report.

Reviewer #3 (Remarks to the Author):

I found the revised version of the manuscript to be a great improvement on an already convincing piece of work. Additionally, I have very much enjoyed reading the response of the authors to the comments of Reviewers 1 and 2.

I strongly support publication of the manuscript in Nature Communications.

REVIEWERS #1' COMMENTS:

Comments on the revised version of “Circular photogalvanic spectroscopy of Rashba splitting in 2D hybrid organic-inorganic perovskite multiple quantum wells” by Xiaojie Liu *et al.*

The authors fully addressed all my comments from the first round of research. I recommend to publish the manuscript in Nature Communications.

Answer: We thank the reviewer for providing very useful comments to help bringing our manuscript to its current level, and are glad to know we have addressed his/her concerns satisfactorily.

Minor corrections:

Ruddleston-Popper → Ruddlesden-Popper

Answer: Correction was made accordingly.

wire grip polarizer → wire grid polarizer

Answer: Correction was made accordingly.

data-points → data points

Answer: Correction was made accordingly.

the slow process such as photothermal effect dominates → slow processes such as the photothermal effect dominate

Answer: Correction was made accordingly.

slight asymmetry among the two Au electrodes → slight asymmetry between the two Au electrodes

Answer: Correction was made accordingly.

due to spin-galvanic effect rather than CPGE → due to the spin-galvanic effect rather than the CPGE

Answer: Correction was made accordingly.

same polarization than → same polarization as

Answer: Correction was made accordingly.

upon absorption by a circular polarized pump → after creation by a circularly polarized pump

Answer: Correction was made accordingly.

Similar situation occurs → A similar situation occurs

Answer: Correction was made accordingly.

from ferromagnet electrode → from a ferromagnetic electrode

Answer: Correction was made accordingly.

Rashba-Edlestein → Rashba-Edelstein

Answer: Correction was made accordingly.

Reviewer #2 (Remarks to the Author):

My opinion is that the manuscript can be published in Nature Communications in the revised form.

In the revised version, the circular reflectivity topic has been removed from the main text. This means that my comment 3 is not related to the revised version. Nevertheless, I would like to state that the authors' answer to the comment 3 is unsatisfactory. Figure R1 in the rebuttal can in fact be applied for a process where the incident particle transforms into two particles. In the reflectivity, one has a transformation of an initial state of a particle (photon) into a final state of a particle (photon). The reflection is an extreme manifestation of the light scattering. The time invariance implies coincidence of the probabilities of the scattering processes $\alpha \rightarrow \beta$ and $T\beta \rightarrow T\alpha$, where α and β are the states participating in the scattering and T is the time reversal which, in addition to changing initial and final states, flips the wave vectors and circular polarization, so that $R(\text{RCP}, \theta) = R(\text{LCP}, -\theta)$, as is written in my previous report.

Answer: We thank the referee for his/her comment. We will consider the referee's point in more detail and will apply the appropriate symmetry analysis to relevant cases.

Reviewer #3 (Remarks to the Author):

I found the revised version of the manuscript to be a great improvement on an already convincing piece of work. Additionally, I have very much enjoyed reading the response of the authors to the comments of Reviewers 1 and 2.

I strongly support publication of the manuscript in Nature Communications.

Answer: We thank the reviewer for providing very useful comments to help bringing our manuscript to its current level, and are glad to know we have addressed his/her concerns satisfactorily.